# A qualitative examination of the factors affecting the adoption of injury focused wearable technologies in recreational runners

**Aisling Lacey**[1,2]*, **Enda Whyte**[1,3], **Sinéad O'Keeffe**[1,3], **Siobhán O'Connor**[1,3], **Kieran Moran**[1,2]

**1** School of Health and Human Performance, Dublin City University, Dublin, Ireland, **2** Insight SFI Research Centre for Data Analytics, Dublin, Ireland, **3** Centre for Injury Prevention and Performance, School of Health and Human Performance, Dublin City University, Dublin, Ireland

* aisling.lacey5@mail.dcu.ie

**Data Availability Statement:** All relevant data are within the paper and its Supporting information files.

## Abstract

### Purpose

Understanding the perceived efficacy and ease of use of technologies will influence initial adoption and sustained utilization. The objectives of this study were to determine the metrics deemed important by runners for monitoring running-related injury (RRI) risk, and identify the facilitators and barriers to their use of injury focused wearable technologies.

### Methods

A qualitative focus group study was undertaken. Nine semi-structured focus groups with male (n = 13) and female (n = 14) recreational runners took place. Focus groups were audio and video recorded, and transcribed verbatim. Transcripts were thematically analysed. A critical friend approach was taken to data coding, and multiple methods of trustworthiness were executed.

### Results

Excessive loading and inadequate recovery were deemed the most important risk factors to monitor for RRI risk. Other important factors included training activities, injury status and history, and running technique. The location and method of attachment of a wearable device, the design of a smartphone application, and receiving useful injury-related information will affect recreational runners' adoption of injury focused technologies.

### Conclusions

Overtraining, training-related and individual-related risk factors are essential metrics that need to be monitored for RRI risk. RRI apps should include the metrics deemed important by runners, once there is supporting evidence-based research. The difficulty and/or ease of use of a device, and receiving useful feedback will influence the adoption of injury focused running technologies. There is a clear willingness from recreational runners to adopt injury focused wearable technologies whilst running.

**Funding:** KM was responsible for funding acquisition for this research project. This publication has emanated from research supported by Science Foundation Ireland (SFI) under Grant Number SFI/12/RC/2289_P2, co-founded by the European Regional Development Fund. Funding for the study was received as part of a large-scale, centre-wide funding from Science Foundation Ireland to develop Insight (the national research centre for data analytics: www.insight-centre.org). The funders had no role in study design, data collection and analysis, decision to publish, or preparation of the manuscript.

**Competing interests:** The authors have declared than no competing interests exist.

## 1. Introduction

Wearable technologies, including mobile phones and smart watches, are devices that can be worn or carried by an individual that can include measurement capabilities used to assess and monitor physical activity, movement, health and well-being [1, 2]. Advancements in wearable technologies have made it possible for the early detection of illnesses and injuries by allowing for continued monitoring of individuals [3]. The use of wearable technologies has become increasingly popular within the running community, with approximately 90% of runners using some form of technology to monitor their training [4]. Primarily, wearable devices in this market function to collect global positioning system (GPS) data and information on running technique to provide summary reports for assisting running performance [5–7]. This is achieved by the tracking of personal running data [8, 9], planning of running goals [10], and/or by increasing a runner's motivation to train [9, 11]. However, despite the high incidence of running related injuries (RRIs), recently reported at 40% [12] and 46% [13], and the popular use of wearable devices to manage other illnesses and injuries [14–16], there is a dearth of research investigating the perceived usefulness of injury focused wearable technologies in runners. Additionally, no effective injury prevention programmes for reducing RRIs have been identified in the literature thus far [17]. It has also been hypothesized that this is due to previous injury prevention programmes focusing on reducing the impact of a single risk factor, when the cause of RRIs is multifactorial [17].

Understanding the underlying factors that drive adoption of wearable technologies is a crucial step in ensuring their successful uptake [18]. One such factor is the perceived usefulness of a device to the user [19, 20]. Adapting the six-stage Translating Research into Injury Prevention Practice (TRIPP) framework [21] to the current context, it is clear that understanding and including the factors contributing to RRI's, while understanding the perceptions and behaviours of potential users in their own sporting context is pivotal in developing a useful device. Therefore, it is crucial to identify the metrics recreational runners perceive as important for monitoring injury risk and adopting injury focused technology.

Identifying runners' perceived facilitators and barriers to the use of wearable technologies is also deemed essential for technology adoption [22]; however, the majority of this research has to date focused on *performance* insights as motivators to the use of wearable technologies [22–26] rather than on injury. Only one study [8] appears to have examined the barriers and facilitators to the use of running technologies for reducing RRIs.

Previous research investigating runners' usage of wearable technologies in relation to performance and injury has predominantly used questionnaires and surveys as the methodological approach [8, 22–25]. To further explore runners' perceptions of such topics, a qualitative study would provide more insightful and detailed understanding [27, 28]. Therefore, the aim of this study was to examine the factors that affect injury focused technology adoption in recreational runners, to identify the metrics perceived as important to monitor for RRI risk, and to identify the perceived facilitators and barriers to the utilization of injury focused technologies.

## 2. Materials & methods

### 2.1. Design

Constructivist grounded theory was deemed an appropriate methodological choice for the current study, as a theory addressing the factors affecting the adoption of injury focused running technologies in recreational runners is yet to be identified. Grounded theory (GT) consists of strategies for developing theories through the analysis of qualitative data [29, 30]. It allows for the investigation of how and why people, communities or organisations experience and

respond to events, challenges and problematic situations [31], and elicits rich, narrative accounts of this experience in order to generate an inductive theory [32]. Constructivist grounded theory (CGT) is similar to GT in the sense that it involves constant comparative analysis and saturation; however, CGT assumes that rather than theories being discovered as in GT, we construct theories through past and present experiences and interactions with people, perspectives and practices [32]. Constructivist grounded theory is an iterative process that follows repeated cycles of data collection and analysis to allow for continuous improvement, expansion and clarity of the emerging theory [33]. There was a need to identify both the perceived facilitators and barriers to adoption as certain factors may act in a bi-directional manner, serving as both facilitators and barriers [34, 35]. Ethical approval was granted by the local university's Ethics Committee. The Standards for Reporting Qualitative Research [36] (S1 Table) were adhered to. A semi-structured focus group schedule was developed by the researchers, and followed an iterative process throughout the pilot study phase (S2 Table).

## 2.2. Participants

A purposive sample of 27 adult recreational runners were recruited. Local running clubs were contacted via email and asked to distribute research information and contact details to potential participants. Those interested then contacted the researchers. Eligible participants had to be aged between 20 and 60 years and meet Mulvad and colleagues' [37] definition of a recreational runner: someone running at least once per week for at least six months. A minimum/maximum running volume or the use of wearable technologies was not included in the inclusion criteria.

## 2.3. Pilot study

A pilot study was conducted to educate and train the primary author in efficient focus group moderation techniques, and in the use of an analytical framework for analysing qualitative data. The results of the pilot study were not included in the main study results. The focus group schedule was updated throughout this pilot phase to include more open-ended questions and additional probes to include all participants in the discussion (e.g., "What do you think M#, have you any thoughts on that?"). Four male and five female recreational runners were recruited as a convenience sample, aged 25.1 years ± 2.2 years. Three separate pilot study focus groups were facilitated by the primary author, each taking place via remote video conferencing software (Zoom, version 5.7.0) and lasted 39.1 minutes ± 5.4 minutes.

## 2.4. Main study procedures

Prior to taking part in a focus group, participants were required to provide informed written consent and complete a short pre-focus group questionnaire. The questionnaire was used to gather demographic information, as well as details on participants' running habits, their usage of running technologies and their experience with RRI's (S1 Appendix). A RRI was defined as any musculoskeletal pain in the lower back/lower limbs that causes a restriction to or stoppage of running for at least seven days or three consecutive scheduled sessions, or that causes a runner to consult a healthcare professional [38]. On completion of the questionnaire, participants were contacted via email to arrange a suitable focus group time. To encourage as much interaction as possible, the focus groups were stratified to include participants of similar age, with similar running backgrounds.

Nine separate focus groups took place with 27 recreational runners (range = 2–4, median = 3 participants per group). Focus groups were moderated by the primary author and lasted 45.1 minutes ± 11.4 minutes. Each focus group began with a brief introduction to the study and the

aims of the focus group were outlined (S2 Table). Participants were encouraged to speak freely and given the opportunity to ask questions throughout. Group discussion began by each participant describing the types of running technologies they use. Following this, a discussion regarding the facilitators and barriers to technology use progressed, with a specific emphasis placed on injury focused running technologies. To aid discussion of injury focused technologies, it was suggested that a hypothetical smartphone application (app) could collect both sensed data (from a sensor) and data that users would be required to input manually (potentially before and/or after a run). Participants were probed to discuss this in relation to their experience with other running- and/or health-related apps, as well as their perceived use of a new injury focused technology (i.e., the hypothetical technology described). Conversation then moved to discuss participants' perceived risk factors for RRIs, and the metrics they deemed important to monitor for RRI risk. On the closing of the focus groups, participants were given another opportunity to ask questions and to provide further comments or statements that they felt may be important. A reflective and iterative approach was taken with regard to focus group moderation and the content of the focus group schedule. Additional probes were included in the focus group schedule and adjustments to moderation techniques, (e.g., ensuring equal speaking opportunities for all participants) were made during this data collection phase.

## 2.5. Data analysis

Frequencies and descriptive statistics were generated from the questionnaire responses using SPSS [version 27.0; IBM Corporation]. Focus groups were audio and video recorded using built in software available in Zoom [version 5.7.0], and transcribed verbatim by the primary author. Participants were allocated an identification number during transcription to maintain anonymity and protect their confidentiality, with responses coded by participant gender (e.g., male = M; female = F). The transcribed focus groups were coded by the primary author using NVivo (QSR International) Constant comparative analysis was conducted, initiated after transcription of the first focus group, and continued throughout the data collection phase [39], and theoretical sampling continued until data saturation was reached [40]. A coding framework was developed and updated by the primary author throughout the data collection phase, and was used in the coding of the transcribed focus groups (S3 Table). Braun and Clarke's [41] methodology for thematic analysis was utilised during data analysis, which involved six key features: familiarisation with the data, generating initial codes, searching for themes, reviewing themes, defining and naming themes, and producing the report [41]. From the identified codes, core categories were identified, with subsequent themes and sub-themes emerging.

## 2.6. Trustworthiness

Multiple methods of trustworthiness were undertaken to ensure the rigorous and accurate presentation of findings. A critical friend approach was used to enhance the analytical process [42], and to establish reliability and ensure rigour of results [43]. The goal of critical friends is not to reach consensus or agree on all aspects of the findings, but rather 'encourage reflexivity by challenging each other's construction of knowledge' [43, 44]. The approach also gives the opportunity for researchers to explore multiple interpretations of the data, reducing the effect of researcher bias [42, 45]. After all transcripts had been coded by the primary author, a percentage of transcripts were coded by an external researcher with experience in qualitative research (SOK). Taking a critical friend approach, researchers (AL and SOK) met on multiple occasions to conduct a coding consistency check on the coded transcripts. Codes, sub-themes, themes and core categories were critically reviewed and discussed. A high level of agreement

was reached, while any disagreements during the analysis were discussed, with varying interpretations presented. This stage of analysis led to the development of some additional codes, as well as the merging of existing codes.

Following this, trustworthiness was further enhanced through investigator triangulation, in which the primary author met with two other members of the research team (KM and EW). Similar approaches were taken to review and discuss the coded data, with any disagreements discussed and appropriate changes made.

Additionally, in the presentation of the representative and accurate findings, multiple examples and direct quotations from transcripts are provided (S4 Table), indicating a broad and diverse contribution from participants during focus groups, reducing the chance of individual bias [46]. Included quotations were agreed upon by researchers.

## 3. Results

Nine focus groups were conducted with 13 (48.1%) male and 14 (51.9%) female recreational runners. Participants were aged 35.0 years ± 10.7 years (range: 23–53 years). Running and injury histories are detailed in Table 1. All participants were currently using, or had done so in the past, at least one form of wearable technology to monitor their running, with GPS watches

**Table 1. Participant running and injury history (n = 27).**

| Running history | | | |
|---|---|---|---|
| Is running your main sport? | *Yes* | *No* | *Unsure* |
| | 63% (n = 17) | 33% (n = 9) | 4% (n = 1) |
| How long have you been running? | *Less than 3 years* | *4–5 years* | *More than 5 years* |
| | 15% (n = 4) | 4% (n = 1) | 82% (n = 22) |
| How often do you run? | *Once a week* | *2–3 times a week* | *4 times a week or more* |
| | 7% (n = 2) | 44% (n = 12) | 48% (n = 13) |
| **Injury history** | | | |
| Have you ever had a RRI? | *Yes* | *No* | |
| | 82% (n = 22) | 19% (n = 5) | |
| Thinking of your worst injury, how much training did you miss? * | *Less than 10 days* | *2–3 weeks* | *4 weeks or more* |
| | 24% (n = 5) | 24% (n = 5) | 52% (n = 11) |
| How many RRI's have you had in the last 12 months? * | *None* | *1 RRI* | *2 RRI's* |
| | 24% (n = 5) | 33% (n = 7) | 43% (n = 9) |
| How important is injury prevention to you? (n = 22) | *Moderately important* | *Very important* | *Extremely important* |
| | 18% (n = 4) | 27% (n = 6) | 55% (n = 12) |
| **Running technology use** | | | |
| *What types of running technologies do you use?* | | | |
| Mobile phone & applications | 48% (n = 13) | | |
| GPS watch | 56% (n = 15) | | |
| Heart rate monitor | 33% (n = 9) | | |
| Smartwatch | 7% (n = 2) | | |
| Wristband activity tracker | 7% (n = 2) | | |
| Body worn sensor | 4% (n = 1) | | |
| Other | 4% (n = 1) | | |

n = number of participants, RRI = running-related injury,

* = missing data

and mobile phones being the most popular devices [used by 55.6% (n = 15) and 48.1% (n = 13) of participants respectively]. Supplementary quotes can be found in S4 Table.

### 3.1. Metrics perceived as important for monitoring RRI risk

Three core categories of risk factors were identified as important for monitoring with injury focused running technologies: overtraining, training-related risk factors, and individual-related risk factors. Within each core category, various themes and sub-themes emerged (Table 2).

**3.1.1. Overtraining.** Excessive loading and inadequate recovery were perceived to contribute to overtraining, and increase an individual's risk for sustaining a RRI. Participants suggested that these factors be monitored by injury focused technologies. Overall, the most common theme emerging from the discussion of risk factors for RRI's was excessive loading. Runners perceived high accumulative loads, high intensity training and in-session fatigue to contribute to excessive loading, increasing the risk for sustaining a RRI (Table 2). Participants perceived that the *"volume of training"* (F6) and *"total mileage"* (M8) are *"big risk factor*[s]*"* (M8) for RRI onset. Additionally, the type and intensity of training, and *"whether you were pushing hard"* (M3) was also perceived to impact the risk of injury; F6—*"The type of running you're doing. If you're doing interval training, long distance, sprints, or the volume of training maybe. . . the impact of that on your injuries"*. Another participant (M2) felt that these factors should be monitored in order to make sure *"the body is able to accumulate those miles"* and how injury focused technologies could function *"to make sure that you're not going into a red zone"* in terms of loading. Some participants also discussed how in-session fatigue can lead to

**Table 2. Running-related injury risk factors perceived as important to monitor using wearable technology devices by recreational runners.**

| Core categories | Themes *(number of participants & focus groups to discuss theme)* | Sub-themes *(number of participants & focus groups to discuss sub-theme)* |
|---|---|---|
| Overtraining | Excessive loading *(17\* participants in 9# focus groups)* | High accumulative load *(12 participants in 7 focus groups)* |
| | | High intensity training *(11 participants in 8 focus groups)* |
| | | In-session fatigue *(5 participants in 5 focus groups)* |
| | Inadequate recovery *(13 participants in 7 focus groups)* | Fatigue & poor sleep *(6 participants in 5 focus groups)* |
| | | Poor nutrition *(6 participants in 4 focus groups)* |
| | | Insufficient rest days *(5 participants in 4 focus groups)* |
| Training-related risk factors | Training activities *(13 participants in 6 focus groups)* | Concurrent training activities *(12 participants in 6 focus groups)* |
| | Running technique *(10 participants in 5 focus groups)* | Foot strike technique *(5 participants in 4 focus groups)* |
| | | Bilateral asymmetry *(4 participants in 3 focus groups)* |
| | | Cadence *(3 participants in 3 focus groups)* |
| | Running environment *(9 participants in 7 focus groups)* | Terrain *(8 participants in 7 focus groups)* |
| | Footwear *(8 participants in 5 focus groups)* | Type of footwear *(6 participants in 4 focus groups)* |
| | | Infrequent changing of footwear *(4 participants in 4 focus groups)* |
| Individual-related risk factors | Injury status & history *(11 participants in 5 focus groups)* | Ongoing niggle *(7 participants in 6 focus groups)* |
| | | Previous injury *(6 participants in 3 focus groups)* |
| | Population characteristics *(5 participants in 3 focus groups)* | Age *(4 participants in 3 focus groups)* |
| | | BMI *(3 participants in 2 focus groups)* |
| | Type of runner *(3 participants in 2 focus groups)* | Preferred distance/event *(3 participants in 2 focus groups)* |

Note: Themes and sub-themes are presented in order of those most frequently discussed.

\* indicates out of 27 participants.

# indicates out of 9 focus groups.

inappropriate running technique, and *"as you go into the longer distances"* (M1), your risk of sustaining an injury increases—M7- *"the more tired I get and if I try and stick to a particular pace, the whole form goes out, and I would think that would lead to more injuries in that regard"*. Inadequate recovery was commonly discussed as a perceived risk factor for developing RRI's (Table 2). With the first sub-theme of fatigue and poor sleep, it was perceived that if *"you're not sleeping properly"* (F1), you are more susceptibility to injury. One participant (F8) described sleep as having a *"huge impact"* on injury risk and if you *"don't get enough sleep… your muscles just don't repair as quick, they don't recover as quick.* Insufficient rest days taken was also perceived to increase injury risk. One participant (F3) described how many runners may be *"over running"* and *"probably are injured because they're not actually taking rest days"*, while also describing the importance of monitoring this to ensure *"they're not over-doing it"*. It was also perceived by some that inadequate nutrition may increase the risk of a RRI, with one participant (F11) suggesting that *"so many people don't fuel themselves properly"* and *"so many runners don't eat enough"*, which was perceived as a *"huge factor"* for injury risk.

**3.1.2. Training-related risk factors.**   Training-related risk factors for RRI onset included: training activities, running technique, running environment, and footwear (Table 2). Other training activities that runners may be participating in were commonly discussed. It was perceived that certain activities may either reduce or increase the likelihood of sustaining a RRI, and that it is *"very important to take into account what other sports they're doing"* (M2). It was suggested that participation in various sports (e.g., Gaelic football, rugby, golf, track events) *"predisposed"* (M2) runners to injury, and that it was *"important to take into account… other sporting activity to see if it's an injury related to running, versus related to something else, or a compound of both"* (M2). Participation in activities such as yoga, strength training and swimming were perceived to reduce the likelihood of injury, and that it would be important to monitor *"what people do outside of running, to make themselves stronger"* (M10). One participant (M3) for example perceived that by *"improving my stretching, by doing yoga"*, it *"makes me less injury prone"*. With running technique, participants suggested that foot strike technique, bilateral asymmetries (i.e., a difference between left and right lower limbs), and cadence may be factors that influence the onset of RRI's. Although unclear as to how these factors may influence RRI risk, participants perceived that they were important metrics to monitor. Some participants felt that monitoring *"foot strike"* (M5), *"asymmetry in the heel strike or ground contact time"* (M13), *"whether you're landing heavier left or right foot"* (M7), or *"stride length"* (M12) and *"cadence"* (M12) would give insight into risk of injury. The terrain on which people ran was commonly perceived as a potential risk factor for injury. There was generally a lack of consensus between participants as to which surfaces posed the greatest risk, despite one participant (M2) describing this metric as *"really important to take into consideration"*. However, this theme was frequently identified as an important metric to monitor. Some participants suggested that *"running up a hill"* (F2), running on *"solid concrete"* (M2), and *"constant running on the roads"* (F7) increased the risk of injury. Runners also perceived their type of footwear, and how the infrequent changing of footwear may be important factors in relation to RRI risk. One participant (M7) described their interest in understanding *"how more injury prone you are, dependent on both the age of the runners* [shoes] *you use, and the different brands of runner* [shoe] *you use"*. Some participants described how they would regularly change their footwear to reduce the risk of injury, and how prolonged use of a single pair of shoes can increase the risk of injury; F11—*"I feel like so many people don't change their runners often enough and I really think that's a huge factor in injuries"*.

**3.1.3. Individual-related risk factors.**   The final core category of risk factors surrounded individual-related risk factors (Table 2). Participants discussed the importance of tracking the ongoing injuries and/or *"niggles"* (F2) that they may have, and how monitoring these may give

further insight into the development or prevention of a more serious RRI. One participant (M7) queried whether *"niggles"* were *"precursors to an injury"* or if they were *"just the little aches and pains that we all get?"*. Some participants also described the impact that previous injuries may have on the risk for further injuries, suggesting they should be monitored by injury focused technologies. One participant (M6) described the relationship between previous injuries and their current running, stating; *"the injuries I have, they're all. . . rugby related and contact related, so I find the issues I have running are probably tied back to the issues that I've had playing rugby"*. In relation to population characteristics, participants generally perceived that older age increased the risk of injury and how *"when you're getting older, you're probably going to get more injury prone"* (M8). A greater body mass index (BMI) was also perceived by some to be a risk factor for injury, as *"the more you weigh. . . the higher your impact forces, and I guess that that will be a straight impact. . . on the risk factors"* (M8). A runner's perception of a run was also perceived to be important for monitoring injury risk, as one participant (F14) described; *"how hard did the run feel. . . were you tired before starting, tired during, tired after"*. Mood and *"feelings"* (M10) were also discussed by some participants, with the perception that they *"play a part in your training"* (M10) and should be monitored. As the final sub-theme, it was perceived that the *"type of runner"* (M8) and differences in preferred running distance may influence susceptibility to injury. It was suggested (M4) that *"different types"* of runners *"would have different injuries"*, and that because of their 'differences', runners *"don't have a lot in common in relation to the type of injuries that [they're] likely to pick up"* (M9).

### 3.2. Facilitators to the use of injury focused running technologies

Ease of use and receiving useful feedback were identified as core categories of facilitators to the use of injury focused running technologies (Table 3).

  **3.2.1. Ease of use.**   Perceived ease of use was the first core category identified, with application design and sensor design emerging as themes (Table 3). In relation to the application design, participants suggested a *"user-friendly system"* (M2) that fitted with their current usage habits would facilitate use. In particular, technologies with quick or *"succinct"* (F3) and *"easy to do"* (M11) input sessions, combined with user-friendly questions (such as *"hit a smiley face or give a rating of one to ten"* [M3] or *"tick the box, rate the scale-type things"* [M9]) would encourage use. Participants suggested that a time requirement of 30 seconds to two minutes would be optimal and facilitate their use. The ability to sync a runner's current applications and technologies with a new device was also suggested by many as a facilitator. This was perceived to reduce the burden placed on users, while optimizing the reception of new and useful data; M3—*"especially if the information is already there, maybe you can get it from Strava and tie it in"*. Participants suggested that being prompted by their smartphone would enhance engagement and facilitate their use of an application; F9 -*"a reminder. . . a notification coming up is really handy, because it's easy to forget"*. It was suggested (M5) that data collected by a wearable device that *"updates automatically"* would be *"great"* as reducing user demand would increase compliance; M5—*"the less that data we have to put in, the better"*. It was also commonly suggested that a system and device that fitted into participants' current technology usage habits would be easily adoptable. One participant suggested that engaging with a new application wouldn't be an issue for them as *"I'd be recording it anyway, so to add in something small, it'd be no problem for me"* (M2), while another suggested that it may become part of their current habits; *"at the end of the training session or running session, I would automatically go to my smartphone, look at the Garmin app"* (F8).

  With regard to sensor design, the location, attachment method, and specifications of the sensor were sub-themes of facilitators identified (Table 3). Although some locations were

**Table 3. Facilitators and barriers to the use of injury-focused wearable technologies.**

| Core Categories | Themes | Sub-themes | Facilitators | | Barriers | |
|---|---|---|---|---|---|---|
| | | | Secondary sub-theme | Tertiary sub-theme | Secondary sub-theme | Tertiary sub-theme |
| Use of a wearable device | Application design | User demand | User-friendly system (22 participants in 9 focus groups) | Quick input session (17 participants in 9 focus groups) | High user input requirement (16 participants in 7 focus groups) | Time consuming (>5 mins) (13 participants in 6 focus groups) |
| | | | | Question format (7 participants in 5 focus groups) | | High quantity of questions (>4 questions) (6 participants in 4 focus groups) |
| | | | | Synced with other applications (7 participants in 5 focus groups) | | Repetitive/Irrelevant data required (6 participants in 3 focus groups) |
| | | | | Notifications reminders (6 participants in 4 focus groups) | | |
| | | | | Automatic downloading of data (5 participants in 3 focus groups) | | |
| | | | Current usage habits (13 participants in 8 focus groups) | Fits with current usage habits (13 participants in 8 focus groups) | | |
| | Sensor design | Location | Lower back (8 participants in 6 focus groups) | Convenient (7 participants in 5 focus groups) | Lower back/Waist (8 participants in 3 focus groups) | Uncomfortable (4 participants in 3 focus groups) |
| | | | Foot/Shoe (8 participants in 5 focus groups) | Convenient (8 participants in 5 focus groups) | Wrist/Arm (3 participants in 2 focus groups) | Not secure (4 participants in 2 focus groups) |
| | | | Wrist/Arm (5 participants in 5 focus groups) | Convenient (5 participants in 5 focus groups) | Obvious/Noticeable to others (3 participants in 2 focus groups) | |
| | | | Chest/Torso (5 participants in 4 focus groups) | Convenient (5 participants in 4 focus groups) | | |
| | | Application method | Discrete (non-specific attachment method) (7 participants in 5 focus groups) | | Uncomfortable/Irritating (non-specific attachment method) (8 participants in 5 focus groups) | |
| | | | Comfortable (non-specific attachment method) (6 participants in 5 focus groups) | | Time consuming set-up (3 participants in 3 focus groups) | |
| | | | Convenient (6 participants in 5 focus groups) | | | |
| | | | Belt mechanism (5 participants in 4 focus groups) | Convenient (3 participants in 3 focus groups) | Belt mechanism (5 participants in 3 focus groups) | Uncomfortable (4 participants in 2 focus groups) |
| | | | Clip mechanism (3 participants in 2 focus groups) | Convenient (3 participants in 2 focus groups) | | |
| | | Specifications of sensor | Small (5 participants in 4 focus groups) | | Bulky (8 participants in 7 focus groups) | |
| | | | Lightweight (5 participants in 4 focus groups) | | Large (3 participants in 2 focus groups) | |
| | | Technical features | Infrequent charging (3 participants in 2 focus groups) | | Frequent charging (3 participants in 2 focus groups) | |
| | Feedback | Feedback received | Injury-related feedback (20 participants in 7 focus groups) | Reduce injury risk (11 participants in 7 focus groups) | | |
| | | | | Monitor rehab from injury (10 participants in 5 focus groups) | | |
| | | | | Understand injury mechanisms (7 participants in 6 focus groups) | | |
| | | | | Advice/Recommendations (6 participants in 3 focus groups) | | |
| | | | | Extend running career (3 participants in 1 focus group) | | |
| | | | Enhanced data (8 participants in 4 focus groups) | Cadence/Stride information (3 participants in 3 focus groups) | | |

deemed more preferable than others, there was a lack of agreement between participants on the most preferable location. Participants suggested that once the location was comfortable, convenient and allowed for the device to be stable, this would facilitate their use. One participant (F11) described their perception of the lower back as a potential location and felt that "*your shorts would hold it in place*" and "*it wouldn't be moving around too much*". Another (M9) participant described the convenience of the foot/shoe as a potential location because "*if it's on my runners. . . I'm much more likely to just leave it there. . . rather than forget about it*", while another participant (F1) felt that the wrist/arm would be suitable as "*you don't want to have something that has to be carried or have to adapt to clothes to take along with you*". Finally, it was also perceived by some that the chest/torso would be suitable as from previous experience, "*I don't notice it's there really*" (M11).

Participants also felt that the attachment method of a sensor may act as a facilitator to device use. Personal preference varied amongst participants, however the overwhelming consensus suggests that a stable, comfortable, discrete and convenient attachment method would facilitate device use. Participants suggested that "*if it fits. . . properly*" (M6), and "*can be easily worn and it's not. . . impacting you in any way*" (F8), and "*as long as it's not a cumbersome thing that's interfering with the running*" (M1), they would have "*no problem wearing it*" (M1). Some participants described their preference for a belt-type attachment method as it was perceived as "*straightforward*" (F8) and "*easy to wear*" (F14), while others suggested that a "*paperclip kind of action*" (F2) application method would be "*easy*" (M2) to attach. Participants discussed the favourability of a "*lightweight*" (M8) and "*unobtrusive*" (M8) device, where "*the smaller* [it was] *the better*" (M10), and how this would facilitate use. Finally, it was suggested that a device with a "*good battery life*" (F1) would enhance user compliance and facilitate device use.

**3.2.2. Receiving useful feedback.**   Participants discussed their willingness to engage with a device (i.e., an application and sensor) should it reduce their risk of sustaining an injury and how potentially beneficial "*a device that you can put in your back pocket that will measure when you're putting your body under a level of stress that is likely to cause an injury*" (M1) could be. Others discussed the commonality of injury and how "*everyone picks up a few niggles a year*" (F11), or how there is "*always that chance that you're going to get injured*" (F1), and their interest in using such a device to reduce this risk; "*I think we've all had our fair share of niggles and injuries that you'd rather not have*" (F1). Others discussed the benefits of a device that could monitor their rehabilitation from injury and potentially provide them with data to explain the mechanics of injury; "*I'm sure often there's obvious reasons that we don't even notice, but sure by having an app you'd be like 'Oh well, I did this, and I did this and I shouldn't have done this'*" (F11). Finally, some described their interest in using technologies if they could prolong their running career "*what would. . . make me. . . able to run for more years without the body failing me*" (M2).

Others described their interest in a device that could provide recommendations for "*preventing the injury developing further*" (F5), or receiving advice on "*whatever you should do*" (F5) to best manage injuries. One final facilitator to encourage use of injury focused technologies was enhanced data that runners could receive. Some participants described the desire for additional data that may give them "*an edge*" (F2) and that could potentially "*improve* [them] *as a runner*" (M2). Participants suggested that receiving data related to performance progressions would facilitate their use, while some expressed their interest in receiving "*the extra thing*" (F1) that they may not be getting with their current devices. Examples included information regarding cadence, stride length, or the "*biomechanics*" (M13) of running technique, while others were interested in "*reaffirming some data that I'm collecting already*" (M13).

### 3.3. Barriers to the use of injury focused running technologies

Difficulty of use and ineffective feedback were identified as core categories of barriers to the use of injury focused running technologies (Table 3).

**3.3.1. Application design.** Participants discussed how the application design could act as a barrier to injury focused technology use, with a high demand on the user serving as a barrier. This was discussed in relation to participants' previous experience with other health- and running-related applications, as well as their perceived behaviour for engaging with a hypothetical injury focused application. Participants considered that this hypothetical app would collect both sensed data (from a sensor) and data that they would be required to input manually (potentially before and/or after a run). A large time requirement for inputting data was identified as a potential barrier to technology use, with M5 suggesting: "*realistically if it'll be any more than a couple minutes and people get bored putting in the data*". Participants discussed their tolerance and willingness to engage with such an application, and it was identified that five minutes was deemed the maximum amount of time runners were willing to spend using an application—M6—"*five minutes probably would be my max*". From previous experience, a requirement to respond to a high quantity of questions (more than four questions) was described as "*onerous*" (F8) therefore identifying a further potential barrier. Questions deemed as irrelevant and repetitive were also described as a barrier with one participant indicating; "*It just gets a bit tedious. . .basically it'd* [wearable wrist-based device monitoring sleep and recovery] *ask you loads of questions, and it's like the same questions over and over*" (M11).

**3.3.2. Sensor design.** The second theme of barriers to the use of injury focused wearable technologies was sensor design. Sub-themes of barriers included: attachment method, location, specifications of the sensor and technical issues (Table 3). Personal preference varied in relation to unfavourable device attachment methods. The general consensus suggested that attachment methods which would "*take too long to get in place*" (F11), required the runner to wear "*some contraption*" (M8), may "*cause any discomfort or blistering*" (M10), or one that was loose-fitting, "*bouncing around*" (F6) or "*going to fall off*" (F6), were potential barriers to use. Differences in the non-preferred locations of a wearable sensor were evident, with some describing the lower back as an undesirable location as it was perceived as uncomfortable or that it may "*rub against your skin and get a bit sore*" (M8). Others suggested that wrist or arm-based sensors would be unsuitable as they "*get annoying after a while*" (M2). Variance in the opinion made it difficult to determine any specific location as a barrier to use; however, the majority agreed that locations perceived as uncomfortable, one's which resulted in excessive movement of the sensor, or were "*very obvious*" (F11) to others would result in reduced compliance, and therefore act as barriers to usage. It was frequently suggested that a "*bulky*" (F9), "*clunky*" (M13) or "*heavy*" (F6) sensor would act as a barrier to technology use, as runners perceived it may "*impact their running*" (F9) and may "*annoy [them] during the run*" (M10). Finally, participants reported that a sensor with a short battery life which would require frequent charging may discourage use as it can "*put me off if the battery is low on it*" (F3).

**3.3.3. Ineffective feedback.** It was also mentioned by some participants that irrelevant or inaccurate data, or what they perceived to be "*too much*" feedback would potentially discourage their use of injury focused technologies. Some participants discussed their perception that ineffective data wasn't "*going to help [them]*" (F1) in their training or recovery from injury.

## 4. Discussion

The main objectives of the current study were to provide a qualitative examination of recreational runners' opinions on: (i) the important metrics to monitor for RRI risk, and (ii) the perceived facilitators and barriers to the use of injury focused running technologies. Overtraining,

training-related, and individual-related risk factors are essential metrics that need to be monitored for RRI risk. Difficulty of use of a device will act as a barrier to the use of injury focused running technologies, while ease of use and receiving useful feedback will act as facilitators. Common themes of facilitators and barriers were identified, implying that many factors can act as facilitators as well as barriers [34]. The findings of the current study are similar to the Technology Acceptance Model (TAM) [19] and the Unified Theory of Acceptance and Use of Technology (UTAUT) [20], which suggest that individuals' use of technology will be influenced by a number of factors. Such factors include: the perceived ease of use and perceived usefulness of a device/app (as suggested by the TAM [19]), and the effort expectancy (which is preceded by ease of use, perceived ease of use, and complexity), performance expectancy (i.e., the degree to which an individual believes a technology will help to improve their injury risk [in the case of the current study]), and behavioural intention to use a device/app (as suggested by the UTAUT [20]). Our findings map to these models and as discussed below, we found that perceived usefulness and/or performance expectancy (i.e., the metrics perceived as important for monitoring RRI risk and feedback received), and perceived ease of use and/or effort expectancy (i.e., difficulty/ease of use) will influence recreational runners' behavioural intentions to use injury focused wearable technologies. App developers (those developing smartphone applications) and technology developers (i.e., those designing wearable devices/sensors) can draw upon these theories and the findings of the current study to design and create injury focused wearable technologies suitable for use by recreational runners.

## 4.1. Metrics important for monitoring RRI risk

The broad range metrics perceived as important to monitor for RRI risk highlights participants' awareness of the multifactorial aetiology associated with RRI's, as shown by multiple systematic reviews [47–50]. Overtraining, consisting of excessive loading and inadequate recovery, was perceived as a leading risk factor for the development of RRI's in the current study, in line with current knowledge about RRIs [51] and similar to the perceptions of recreational runners in previous studies [28, 52]. Also similar to the findings of Clermont and colleagues [8], the current participants identified longer distances and higher intensity sessions to be important metrics to monitor for excessive load, and subsequent injury risk. Additionally, inadequate recovery (which included the sub-themes of fatigue and poor sleep, insufficient rest days, and poor nutrition) was also perceived to contribute to overtraining. As in similar research, the importance of sleep and food intake for injury prevention have previously been reported by recreational runners [8]. Overtraining, as reported by participants of the current study, also maps to the biomechanical model of injury, whereby loading of tissues beyond their adaptive capability, combined with insufficient recovery, results in injury [53, 54].

Participants identified the importance of monitoring certain training-related metrics for risk of RRI's. Terrain received significant attention as an important risk factor to monitor. While some perceived harder terrains to increase the risk of injury, there was a lack of consensus as to which type of terrain poses greater risks. Harder terrains with less deformation have been hypothesized to result in higher impact forces, increasing the risk of injury [55, 56]. However, while some individual studies have found harder surfaces to produce higher loading [55, 57, 58], other studies have not [56, 59]. Previous systematic reviews [50, 60] have not found terrain to be a significant risk factor for injury. Our participants perceived that participation in other sports (such as rugby, Gaelic football, golf and track events), increased a runner's risk of RRIs. While it has been suggested that additional participation in other sports adds to the cumulative stress placed on the body, increasing the risk of injury [61], a prospective study found that increased weekly volume of other sport participation (i.e., concurrent training)

reduced the risk of RRI's [62]. With running technique, it was suggested that foot strike technique, cadence, and bilateral asymmetry were important to monitor, although participants did not describe how these factors influenced RRI risk. In a similar study, certain aspects of running technique (such as joint motion, ground contact time, and centre of mass motion) were the lowest ranked metrics by participants amongst a list of factors presented to them by the authors as potentially preventing RRI's [8]. Systematic reviews and meta-analyses have been unable to identify strong justifications for the role of specific biomechanical risk factors in the onset of RRI's [63, 64]. While foot strike technique has been suggested to be causative of RRI's based on the increased load that some techniques produce (especially rear-foot strike [65, 66]), a systematic review concluded that there is very low evidence to suggest a relationship with RRI's in general [67]. In relation to increased cadence, while a systematic review found that increasing cadence reduces the magnitude of key biomechanical factors (such as joint kinematics and kinetics, and whole body loading) associated with RRI's [68], a recent systematic review and meta-analysis concluded that average cadence does not differ between injured and uninjured runners [69]. Bilateral asymmetry, which relates to differences between the left and right lower limbs, has been suggested as a risk factor for RRI's based on the premise that because one leg is subjected to more loading, it is predisposed to injury [70, 71]. Again the literature is contrasting, with some studies finding significant limb asymmetries in injured runners both retrospectively [72] and prospectively [73] compared to uninjured runners, while some studies report no differences in asymmetry [70, 74]. No systematic review drawing an overall conclusion has been published to date. Footwear was the final sub-theme of training-related metrics identified, with perceptions that older shoe age increased injury risk. This perception may be associated with the theory that shoe cushioning decreases loading on the body [75, 76], and a decrease in cushioning capacity with extended use increases the risk of RRI's [77, 78]. However, a recent systematic review concluded that no evidence-based recommendations could be made for shoe age and preventing RRI's [79].

The final core category identified as important for monitoring RRI risk was individual-related factors. Ongoing 'niggles' were suggested as an important risk factor for RRI onset in the current study. Different from an injury, in which a runner is forced to reduce or stop training for a period of time [38], our participants' perception of 'niggles' is similar to previous research where runners described 'complaints' as 'small pains' with which they can continue to run [28]. Interestingly, previous injury was only discussed in one third of focus groups, despite being found to be the strongest risk factor for further RRI's in a recent systematic review [50]. Runners have also failed to acknowledge the importance of a previous injury as a risk factor for injury in an earlier qualitative study [52]. While this may reflect a sense of being 'unable to change' the occurrence of having a previous injury, it clearly should be taken into account (via an application) when monitoring for the purpose of preventing re-injury. Population characteristics, including age and BMI, were mentioned by some participants in the current study. It was perceived that older age and greater BMI increased the risk of RRI; however a recent systematic review found conflicting and inconsistent findings for both age and BMI as a risk factor for RRI in short and long-distance runners [50].

It is also important to note that some risk factors for RRI's were not mentioned in the current study, despite being shown as potential risk factors in the literature. For example, sex was not mentioned but has received some attention in the literature. Although findings are mixed, systematic reviews have reported males [50, 80] and females [81, 82] to be at a greater risk for specific RRI's. Additionally, monitoring ground reaction forces (peak and rate) as an indication of how hard someone strikes the ground was not mentioned by participants in the current study, but previous systematic reviews [83] and meta-analyses [63, 76] have investigated the relationship to RRI risk. While there are 'conflicting' [63] and 'inconsistent' [64] results for a

relationship with RRI's in general, high peak and rates of loading have been found to contribute to the development of *specific* RRI's, such as bony and soft tissue injuries [75] and stress fractures [76].

These findings also raise the question about how runners form their opinions that a metric is a risk factor for RRIs, when the research evidence would suggest it is not a risk factor. These perceptions may be due to widely available information on popular running websites. There are many examples of low cadence [84], heel-striking [85], and harder terrains such as concrete [86] being described as risk factors for RRI's on websites, despite a lack of supporting scientific evidence. Clearly there is a need for the scientific community to better educate runners.

These findings expand on the current evidence and report new findings in relation to the metrics deemed important by runners for monitoring RRI risk when using wearable technologies. Firstly, injury focused technologies should monitor risk factors that are deemed important by runners, where evidence-based research supports their relevance (e.g. excessive loading and inadequate recovery). The challenge for app developers is whether to include metrics that monitor risk factors that are: (i) not deemed important by runners, but research does support their relevance (e.g. previous injury), or/and (ii) that are deemed important by runners, but current research does not support their relevance (e.g. terrain and foot strike technique). In the case of the first point, the authors would strongly advocate for including factors supported by evidence-based research (e.g., previous injury), with efforts made by app developers to educate runners in potentially valuable metrics. This is important in order to improve the perceived usefulness of devices [19, 20, 21]. In the case of the second point, the inclusion of these metrics (e.g., terrain and foot strike technique) may be useful if they encourage technology adoption and uptake. This must be balanced against the additional time needed by the user to input this data, which itself may be a barrier to app adoption and continued utilization (discussed below). Also, a lack of research evidence (or mixed evidence) to support a relationship between a metric and an increased risk of a RRI does not necessarily indicate that there is no relationship, but may more reflect the inability of current research to effectively examine the relationship. For example, examining the relationship between running impact loading and injury has been predominantly limited to a one-off assessment, frequently in a laboratory environment [87]. Further research is required to further support the perceived usefulness of metrics that are not currently evidence based but were deemed important by runners of the current study. In addition, future research should include clinicians and running coaches, as their thoughts and opinions may yield further insight into the metrics deemed important for monitoring RRI risk. Development of an app which incorporates a wearable sensor (e.g. an accelerometer) to monitor impact loading and collect user input data on injury status would allow long-term and ongoing monitoring of runners in their natural environment. This would provide more precise and ecologically valid data to better explore whether a relationship does exist. The above findings are also important to coaches and clinicians in developing intervention strategies for injury prevention, where uptake and adherence by runners is improved when runner perception aligns with intervention design [21].

## 4.2. Difficulty/Ease of use

The first identified core category of both facilitators and barriers was in relation to the perceived difficulty and ease of use of injury focused technologies. A table of recommendations for the design focused smartphone app and wearable sensor is provided in the Supporting Information (S5 Table).

**4.2.1. Device design.** Participants indicated that excessive device weight and size are potential barriers to technology use, with unobtrusive and comfortable devices facilitating use.

They also suggested that the attachment method of a device could act as a potential barrier and/or facilitator to use. Varied preferences existed, however the overwhelming consensus suggested that if a device caused irritation or was excessively mobile on the body and interfered with running, this would act as a barrier to use; while a device that was stable and discrete would facilitate use. These perceptions align with previous findings for comfort [88–92], obtrusiveness [91, 92] and device aesthetics [92] in wearable technologies in general.

One sub-theme which generated a large amount of discussion was where the device was to be worn (wear-location); however no one location dominated as either a barrier or facilitator. For example, some participants perceived the foot or shoe to be a highly suitable location (a facilitator), while others perceived this location to be very unsuitable (a barrier). To the best of the authors' knowledge, sensor location has not been previously investigated in runners. However, it has been suggested that athletes of varying sports (e.g., volleyball) may find wear-location to be a potential barrier to use [93]. Additionally, some participants suggested that they would not like a device to be noticeable or obvious to others, as they did not want others to know they would be self-monitoring. This finding has not previously been identified in recreational runners, but has been found in relation to health based monitoring with wearables [94]. Therefore, we suggest a device that could be worn on a variety of locations without negatively impacting on the accuracy of the captured information. Finally, a device with a short battery life was identified as a further barrier to technology use, in line with previous studies on wearable devices [88, 91, 95].

**4.2.2. Application design.** Participants reported that their use of a device would be positively influenced by a user-friendly system, with minimal user input requirement, in line with previous findings for sport tracking technologies [93]. Our participants suggested that as the time requirement and manual input demand to engage with an application increased, their interest and tolerance to engage would decrease. Additionally, it was found that the format of questions within an application could influence compliance. Questions requiring a high amount of text input would discourage engagement, whereas questions formatted visually, with a quick response-time (e.g., tick-the-box) would encourage engagement. These findings have been reported in previous research for users of a weight-loss application [96], and an athlete self-reported measure (monitoring metrics including training, well-being, injury and nutrition) [97].

It was identified that if the use of an injury focused device could conform with participants' current usage habits, it would also facilitate use. Similarly, easily integrating new technologies with existing routines, and the absence of a need for behavioural change has been reported as means of enhancing technology adoption [98, 99]. Compatibility between participants' current wearable devices and/or monitoring applications and a new injury focused device was also identified as a facilitator. Our participants perceived that this would reduce the manual input demand on the user, and result in more accurate and useful information; factors which have been found to enhance wearable technology use [26, 88, 90, 97, 99]. This is important as minimising burden and maximising interest in users leads to improved initial and sustained device compliance [97].

## 4.3. Receiving useful feedback

One final core category of facilitators identified was receiving useful feedback. Receiving relevant, useful and accurate data regarding RRI risk was identified as a facilitator, with participants describing their desire for feedback that could reduce their injury risk, monitor their rehabilitation from injury, and help them understand the mechanisms of injury. It is well understood that maintaining user interest [95, 99] and receiving useful and accurate data [90,

99] can facilitate the use of wearable technologies; while the collection and reporting of inaccurate data and irrelevant information have been suggested as barriers to use of physical activity tracking technologies [26, 88–91, 93].

In line with the TRIPP model (Stage 5: intervention context to inform implementation strategies), the successful implementation of injury prevention practices will be determined by, among other factors, the likelihood of its uptake [21]. In order to improve uptake, researchers (and those issuing injury prevention programmes) must understand why injury prevention practices may or may not be adopted by the target population and provide confidence that adoption of the intervention will reduce the likelihood of injury [21]. Additionally, the TAM [19] and UTUAT [20] models suggest that technologies are more likely to be adopted if they are perceived as both useful and easy to use. Receiving relevant feedback is one way of improving the perceived usefulness of a device, as suggested by participants of the current study.

Some participants also suggested that receiving enhanced data, specifically related to running performance, beyond what they are currently collecting would facilitate their use of injury focused technologies. In the interest of developing a useful injury focused device, these findings are particularly beneficial as they may help to improve perceived usefulness, and ultimately adoption and usage behaviour.

## 5. Strengths and limitations

The current study provides insight into the factors affecting the adoption of injury focused technologies in recreational runners. A representative sample was included, gathering the perceptions of runners of various ages and running backgrounds. Employing constant comparative analysis throughout the data collection phase improved the study's methodological rigour. Furthermore, during data analysis, the involvement of multiple coders with different research and lifestyle backgrounds reduced the impact of potential researcher biases on the interpretation of findings, enhancing the credibility of results.

Although all participants in the current study had used at least one form of wearable technology to monitor their running, bringing valuable experiences in the formation of opinions; the authors believe that the thoughts and opinions of non-users, and those who stopped using wearable technologies are equally as valuable, and should be included in further research. Participants were recruited from Irish running clubs, and therefore findings may not accurately represent the opinions of the global population of recreational runners. The current study did not stratify participants into 'type of runner' (e.g., casual, social or competitive) as in previous studies of recreational runners [8, 22]. Variance in opinion may potentially exist between types of recreational runner, and to examine this could yield further insights into the means of enhancing compliance. Finally, there was potential scope for additional probing during the data collection phase, with some topics requiring further exploration and explanation. For example, runners' perceptions of including an 'overall RRI risk score' into wearable technologies was not examined in the current study. This additional probing and line of questioning may potentially yield further information; an observation that should be considered by future researchers.

## 6. Conclusion

Overtraining, training-related, and individual-related risk factors are essential metrics that need to be monitored using wearable technologies for RRI risk. Some of the metrics valued by participants are supported by scientific evidence (e.g., excessive loading and inadequate recovery); however, they also identified factors that are not clearly supported by scientific evidence (e.g., terrain and foot strike technique), and placed less importance on some factors that are

more strongly supported by scientific evidence (e.g., previous injury). Technology developers should include metrics deemed important by runners, once there is supporting evidence-based research. They should consider the impact of the inclusion of any additional metrics (i.e., those perceived as useful but not supported by evidence, and those supported by evidence but not perceived as useful) and their effect on sensor wearability and excessive user input requirement. Additionally, it would be interesting to investigate the thoughts and perceptions of running coaches and clinicians on the important metrics to include for reducing RRI risk. Difficulty of use of a device will act as a barrier to the use of injury focused running technologies, while ease of use and receiving useful feedback will act as facilitators. To further enhance user compliance, the authors suggest technology developers design an unobtrusive, discrete and comfortable device, designed with a user-friendly system. Findings suggest that if individual users could dictate device location and attachment method, without affecting the accuracy of the technology to monitor risk of injury, this would address these barriers. Preference was given to devices that would also provide runners with information on reducing their individual injury risk, monitor rehabilitation from injury, and provide insight into the mechanisms of injury. Overall, there is a clear willingness from recreational runners to adopt an injury focused wearable device whilst running.

## Supporting information

**S1 Table. Standard for Reporting Qualitative Research (SRQR) checklist [41].**
(DOCX)

**S2 Table. Focus group introduction, aims, schedule domains, and sample questions.**
(DOCX)

**S3 Table. Coding framework.**
(DOCX)

**S4 Table. Supplementary quotes.**
(DOCX)

**S5 Table. Recommendations for the design of an injury focused smartphone application and wearable sensor.**
(DOCX)

**S1 Appendix. Pre-focus group questionnaire.**
(DOCX)

## Acknowledgments

The authors would like to thank the focus groups participants for their participation.

## Author Contributions

**Conceptualization:** Aisling Lacey, Enda Whyte, Kieran Moran.

**Data curation:** Aisling Lacey, Enda Whyte, Kieran Moran.

**Formal analysis:** Aisling Lacey, Enda Whyte, Sinéad O'Keeffe, Kieran Moran.

**Funding acquisition:** Kieran Moran.

**Investigation:** Aisling Lacey.

**Methodology:** Aisling Lacey, Enda Whyte, Sinéad O'Keeffe, Kieran Moran.

**Supervision:** Enda Whyte, Kieran Moran.

**Visualization:** Aisling Lacey, Enda Whyte, Sinéad O'Keeffe, Siobhán O'Connor, Kieran Moran.

**Writing – original draft:** Aisling Lacey, Enda Whyte, Kieran Moran.

**Writing – review & editing:** Aisling Lacey, Enda Whyte, Sinéad O'Keeffe, Siobhán O'Connor, Kieran Moran.

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
