## [Decision Letter · Decision Letter 0]

4 Apr 2022

PONE-D-22-06139A qualitative examination of the factors affecting the adoption of injury focused wearable technologies in recreational runnersPLOS ONE

Dear Dr. Lacey,

Thank you for submitting your manuscript to PLOS ONE. After careful consideration, we feel that it has merit but does not fully meet PLOS ONE’s publication criteria as it currently stands. Therefore, we invite you to submit a revised version of the manuscript that addresses the points raised during the review process, that concerns mainly in a refinement of the manuscript (reoarganisation, expanding some parts).

We look forward to receiving your revised manuscript.

Kind regards,

Laurent Mourot

Academic Editor

PLOS ONE

Journal Requirements:

a) Did participants provide their written or verbal informed consent to participate in this study?

[The authors would like to thank the focus groups participants for their participation. This publication has emanated from research supported by Science Foundation Ireland (SFI) under Grant Number SFI/12/RC/2289_P2, co-founded by the European Regional Development Fund.]

 [KM was responsible for funding acquisition for this research project. This publication has emanated from research supported by Science Foundation Ireland (SFI) under Grant Number SFI/12/RC/2289_P2, co-founded by the European Regional Development Fund. Funding for the study was received as part of a large-scale, centre-wide funding from Science Foundation Ireland to develop Insight (the national research centre for data analytics: www.insight-centre.org). The funders had no role in study design, data collection and analysis, decision to publish, or preparation of the manuscript.]

Reviewers' comments:

Reviewer's Responses to Questions

**Comments to the Author**

1. Is the manuscript technically sound, and do the data support the conclusions?

Reviewer #1: Yes

Reviewer #2: Partly

Reviewer #3: Yes

2. Has the statistical analysis been performed appropriately and rigorously? 

Reviewer #1: N/A

Reviewer #2: Yes

Reviewer #3: Yes

3. Have the authors made all data underlying the findings in their manuscript fully available?

Reviewer #1: Yes

Reviewer #2: Yes

Reviewer #3: Yes

4. Is the manuscript presented in an intelligible fashion and written in standard English?

Reviewer #1: Yes

Reviewer #2: Yes

Reviewer #3: Yes

5. Review Comments to the Author

Reviewer #1: PONE-D-22-06139 _ A qualitative examination of the factors affecting the adoption of injury focused wearable technologies in recreational runners

Thank you for the opportunity to review this article:

General comments:

The facilitators section read much more fluidly than the barriers sections. I would suggest moving this section first and having the barriers come after. The reason being, that there are much more details in the facilitators sections than the barriers. Additionally, it seems that the facilitators are more impactful than the barriers and since all of them use the equipment, it would be good to know what they like about it first.

Generally, it is not clear what sort of “device” the runners are discussing in terms of monitoring injury. Is this just a hypothetical app or something that they already know about/use? It is also unclear who the “manufacturers” are. I would guess that most app developers do not consider themselves as manufactures.

No need to use decimals among the percentages. Does not add to anything by specifying the decimal.

A simplified table of the most important factors (ranked) would be useful. And maybe a table of recommendations for app/device developers. It would be good to know how many metrics should be included in a device/app. There seems to be some categories (i.e. metrics, wearability, usability etc) that could be explained a bit better…again, potentially with a table or figure of some kind.

The discussion section could be improved with some further headings to structure and enhance flow. There is also a lot of repetition of study results in the discussion section. Suggest revising whole section to improve clarity/flow. The superfluous word-padding can be clarified by writing in an active voice and removing some of the nominalisations that are found throughout the whole paper.

Specific Comments:

Please see my comments with associated line numbers throughout your document.

Abstract:

Line 20: the word “use” is presented 4 times in this sentence…rephrase to improve flow.

Line 32: Unclear what you mean that the attachment method and app are barriers and facilitators.

Introduction:

Line 44-46: by “accurate and objective” does this mean valid and reliable?

Line 49: specify “data”

Line 62 - 63: rephrase to active voice improve clarity “Therefore, it is crucial to identify the metrics rec. runners perceive as important for monitoring injury risk and adopting injury-prevention technology”.

Line 69: remove sentence as it is does not add to the argument and comes across as arrogant.

Line 74 - 75: rephrase to active voice to improve clarity (remove nominalised verbs)

Methods:

Did you use the COREQ guidelines (or similar?) for reporting your study? State this at the front of your methods.

Line 104: spell out the number “6”

Lines 107 – 109: rephrase to active voice to improve clarity.

Line 121: spell out numbers zero to nine…correct throughout

Lines 139-141: Please clarify this process…do not understand what you mean by success and success of each discussion topic.

Line 165: was reliability of coding reported?

Line 174: if reliability is not reported in the results, could be stated here.

Results:

Table 1 is difficult to follow…maybe a graph would be easier? I’m not sure I have a suggestion to improve, it’s just hard to follow.

Line 245: remove contraction “haven’t”…not part of the quote so should be spelled out.

Lines 255 and 258: repetitive statements here…one is the quote and one is the authors statement. Rephrase these sentences to improve flow

Line 262: include [shoes] next to the participants use of word “runners” as it could be confusing to the reader and interpreted as a person rather than the shoes.

Line 295: The category “useless feedback” seems a bit harsh and inaccurate considering the subthemes. Too much feedback might not be useless and the feedback delivery subtheme does not fit under the main theme…suggest a main theme along the lines of “ineffective information” or “poor knowledge translation”.

Line 301-311: This section is not very clear. It would be more beneficial to describe specific apps (without naming??) that participants discussed and then talk about those barriers. Apps that I would expect most runners use, especially if they are paired to a smart watch, often collect the data automatically. So I guess I am not aware of what functions apps have where a lot of repetitive questions are asked….Furthermore, did you ask about what types of equipment they use? This would be a good thing to report in a participant characteristics table. Then the reader could make some judgements about the equipment discussed in this section (and I’m assuming further sections).

Line 321: should be “non-preferred”

Line 326: second time using the phrase “general consensus” in this paragraph. Need to rephrase some of the sentences to improve flow.

Discussion:

Lines 441 – 443: Unclear what you mean here.

Line 454: new paragraph, have switched topics from terrain to cross-training

Line 461: new paragraph, have switched topics from concurrent sports to running technique.

Line 476: explain what you mean by ‘bilateral asymmetry’

Line 481: new paragraph

Line 492-93: please clarify this sentence. Hard to follow your argument here

Line 494: who should be monitoring the runner’s injury history?

Line 499: improve sentence structure so that you do not end the sentence with a preposition

Lines 513-514: unclear what you mean by “general” and “specific” RRIs

Line 516: remove the word “both”

Line 518: remove the word “clearly”

Line 519 – 524: I struggle to agree with your statement here. You have just spent a great deal of effort identifying that there is not a lot of evidence-based support in favour of this study’s participant’s ideas/thoughts. You then follow on to leave it up to app developers to decide what should be provided. This seems counter-productive for the purpose of this study. It would make a stronger argument to just state that if manufacturers decide to develop products that focus on the runner’s wants/needs…then do XYZ…and be specific rather than just state “include factors supported by evidence-based research”…State that apps need to be designed that include specific components, and that more research is needed to further support the factors that are not currently evidence based, but are deemed important by runners.

Line 533: difficult to follow with double negative statement

Lines 542 – 545: long topic sentence. Improve for clarity and flow.

Lines 545- 550: sentences here are not supported with the topic sentence…need to create a new paragraph for these and add content for first sentence.

Line 573: please clarify “device location” do you mean “wear-location” as explained above? Be consistent with terms here.

Line 575: improve sentence clarity “to be seen to be….” Hard to follow

Line 612: it is still unclear what “useless information” means

Line 613 – 616: this argument is not well developed. I think this could be improved by explaining which components of the models you are trying to link together here…maybe in a new paragraph and some supporting evidence.

Line 623: improve sentence flow “qualitative informative insight” is a lot of words when you mean “perceptions”

Line 627: the comparative analysis is a strength of rigour during data analysis…be specific about the strengths of your study

Line 645 – 646: this is a strong statement that should be presented near the front of the discussion.

Reviewer #2: PLOS ONE

A qualitative examination of the factors affecting the adoption of injury focused wearable technologies in recreational runners

Thank you for the opportunity to review this manuscript. This qualitative paper aims to identify what is important to runners for monitoring running related injury risk and to identify the barriers and facilitators to the use of injury focused wearable technologies. The paper is well written overall and the methods are very strong, however I do think some significant improvements could be made to the results and discussion. I believe this would improve the impact of the paper to make more concrete changes in the field. I hope the authors find my comments useful.

Major Comments:

1. Supplementary quotes to build trustworthiness: I appreciate there are word count challenges with qualitative results, however I did find that many of the quotes included in the results section were very short, sometimes even only 1-2 words. I would suggest including a supplementary table that includes additional quotes representing each sub-theme. This will grow the readers trust in your findings. The way the results are currently presented the data (quotes) rely heavily on the authors text to be interpreted.

*** Note, that now I am seeing that there are some further data included in the Supplementary material E- this was not referenced when presenting the results, this needs to be flagged for the reading. I would suggest minimum 3 quotes per subtheme, most only appear to have 1.

2. Further refinement of themes: In table 2,3, and 4 I found it interesting that the authors choose to include the number of participants who discussed each theme. While this is good for transparency it also made me think that some of the results needed further consideration – for example subthemes in table 2 high stress and weather only were mentioned by 1 participant. Does that truly warrant them to be their own subtheme? Perhaps there needs to be further refinement of subthemes? Similarly, I noticed this in table 3 and 4 with regards to barriers and facilitators. The authors mentioned how there are many factors that can be both barriers and facilitators. I’m wondering if there is a better way to present this data showing how certain factors can be both. Maybe you could consider combining these tables/themes and I think this could be more meaningful for the reader and future work. For example, instead of having 2 separate core categories as “useless feedback received’ and “receiving useful feedback” consider 1 core category “feedback” then show the useful and useless side by side in same table.

3. Discussion needs to be more focused and concise. There is a lot of repeating of the results in the discussion, I would try to limit this. For example, the discussion repeats results on what factors runners believe are important for monitoring. It would be nice if authors can instead of repeating results, maybe suggest how these factors can actually be monitored? Or what research has been previously been done to monitor these risk factors. For example, I know my garmin watch does track some of these overtraining themes. (line 550).

Also there seems to be a lot of discussion about runners perceived risk factors for injury compared to what the evidence says for risk factors for injury. This seems to be a bit off topic from the original aims of the study. Suggest shortening some of these paragraphs and make sure it stays focused on main purpose of this study.

The concluding paragraph is very strong! Use this to guide the discussion – expand on ideas brought up here.

4. Looking at the supplementary files of the pre-focus group questionnaire, there is a lot of valuable descriptive data that was collected that could be used be better describe the sample for the reader. This could be included in Table 1 and/or at the strart of the results. Details such as do you take part in organized running events, preferred running distance, average training mileages, social running context, etc.

Introduction:

1. Line 51-52: the authors state there is a high rate of RRIs – perhaps a sentence or two and reference that provides some details about RRI incidence in recreational runners.

2. Line 53-54: While the authors state wearables have not been investigated for usefulness in injury prevention, if they could touch on what other injury prevention strategies that have been researched – and the fact that they are not very successful overall – would strengthen the introduction.

Methods:

1. Lines 101-104: suggest including more clear inclusion criteria in the methods and details about how these recreational runners were purposively sampled – i.e. running clubs, online, social networks? I noticed in the limitations it mentioned that running volume, etc was not considered in the sampling strategy. This could be mentioned here. In addition, I would move the details about actual number recruited and their demographics to the beginning of the results section.

2. Line 107-113: Pilot study results. It is not clear if the focus groups/ analysis here were included in the main study results or if this was only used as training for the primary author. Where modifications made to the workshop guide based on these? If so, please provide some examples.

Results:

See major comments above.

Discussion:

See major comments above.

1. Line 424-426- It sounds like the findings about perceived ease of use and perceived usefulness are already known from previous theories, based on the way this sentence is phrased. Consider explaining how your findings can be interpreted or enhanced using the models.

Table 1:

1. Suggest having n=27 in Title at top of table. Then in the 3 lines with missing data indicate with symbol and footnote for number of respondents.

2. Did no participants have more than 2 RRI’s or is this column 2 or more RRIs?

Table 2 & 3 & 4:

See major comments.

Supplementary material C:

Appears to be a duplicate of tables? Assuming this is the coding matrix that was inductively generated from the data, it is already in the paper?

Reviewer #3: This is an interesting study, which aimed to determine from interviews the metrics deemed important by runners for monitoring running-related injury (RRI) risk, and identify the barriers and facilitators to their use of injury focused wearable technologies. This topic is of utmost importance, given the incidence of RRI (also found in the sample studied) and the increasing availability of connected tools allowing to gather loads of running related metrics , and personal data.

The manuscript is a bit long, but due to the qualitative nature of the studies participants quotes illustrate well the main themes gathered in the tables

The synthesis of the interview gives a good overview of participants opinion, and I think authors should highlight that training load related data and recovery data reported are in line with the current knowledge about RRI. However, I am a bit surprised to see runners running less than 1/week included, this should be justified.

The discussion is good and meaningfull, and as stated lines 542-546, I suggest to write in the conclusion/perspectives that it would be interesting to compare those data to those collected in a sample of coaches/clinician.

Finally, I feel that the section about feebacks could be more meaningfull: would participant be willing to get a “ global” RRI score, or specific advices , ie : load is too much, pay attention to sleep, running biomechanics is altered…

6. PLOS authors have the option to publish the peer review history of their article (what does this mean?). If published, this will include your full peer review and any attached files.

Reviewer #1: **Yes: **Codi Ramsey

Reviewer #2: No

Reviewer #3: No

---

## [Author Response · Author response to Decision Letter 0]

16 May 2022

Thank you for your comments and recommendations. We have edited our manuscript in accordance with these comments.

---

## [Decision Letter · Decision Letter 1]

13 Jun 2022

A qualitative examination of the factors affecting the adoption of injury focused wearable technologies in recreational runners

PONE-D-22-06139R1

Dear Dr. Lacey,

We’re pleased to inform you that your manuscript has been judged scientifically suitable for publication and will be formally accepted for publication once it meets all outstanding technical requirements.

Kind regards,

Laurent Mourot

Section Editor

PLOS ONE

Additional Editor Comments (optional):

Reviewers' comments:

Reviewer's Responses to Questions

**Comments to the Author**

1. If the authors have adequately addressed your comments raised in a previous round of review and you feel that this manuscript is now acceptable for publication, you may indicate that here to bypass the “Comments to the Author” section, enter your conflict of interest statement in the “Confidential to Editor” section, and submit your "Accept" recommendation.

Reviewer #1: All comments have been addressed

Reviewer #3: All comments have been addressed

2. Is the manuscript technically sound, and do the data support the conclusions?

Reviewer #1: Yes

Reviewer #3: Yes

3. Has the statistical analysis been performed appropriately and rigorously? 

Reviewer #1: N/A

Reviewer #3: Yes

4. Have the authors made all data underlying the findings in their manuscript fully available?

Reviewer #1: Yes

Reviewer #3: Yes

5. Is the manuscript presented in an intelligible fashion and written in standard English?

Reviewer #1: Yes

Reviewer #3: Yes

6. Review Comments to the Author

Reviewer #1: Thank you for considering the suggestions made by the reviewers, this manuscript is in a better state now.

Well done.

Reviewer #3: The authors adressed my main concerns of the first review. The manuscript gives now more useful insights and is easier to read, even if by nature still long as it containes lots of qualitativs data. It should constitute a nice base for future researches

7. PLOS authors have the option to publish the peer review history of their article (what does this mean?). If published, this will include your full peer review and any attached files.

Reviewer #1: **Yes: **Codi Ramsey

Reviewer #3: No

---

## [Editor Report · Acceptance letter]

15 Jun 2022

PONE-D-22-06139R1 

A qualitative examination of the factors affecting the adoption of injury focused wearable technologies in recreational runners 

Dear Dr. Lacey:

I'm pleased to inform you that your manuscript has been deemed suitable for publication in PLOS ONE. Congratulations! Your manuscript is now with our production department. 

Kind regards, 

on behalf of

Dr Laurent Mourot 

Section Editor

PLOS ONE